# Neural Network Prediction for Ice Shapes on Airfoils Using iceFoam Simulations

**Sergei Strijhak** [1,2,*], **Daniil Ryazanov** [1], **Konstantin Koshelev** [1,3] **and Aleksandr Ivanov** [1,4]

1   Ivannikov Institute for System Programming of the Russian Academy of Sciences, Alexander Solzhenitsyn St. 25, 109004 Moscow, Russia; ryazanov@ispras.ru (D.R.); koshelevkb@mail.ru (K.K.); av.ivanov@ispras.ru (A.I.)
2   Moscow Aviation Institute, Volokolamskoe Shosse 4, 125993 Moscow, Russia
3   Institute for Water and Environmental Problems, Siberian Branch of the Russian Academy of Sciences, Molodezhnaya St. 1, 656038 Barnaul, Russia
4   Keldysh Institute of Applied Mathematics of the Russian Academy of Sciences, Miusskaya Sq. 4, 125047 Moscow, Russia
*   Correspondence: s.strijhak@ispras.ru

**Abstract:** In this article the procedure and method for the ice accretion prediction for different airfoils using artificial neural networks (ANNs) are discussed. A dataset for the neural network is based on the numerical experiment results—obtained through iceFoam solver—with four airfoils (NACA0012, General Aviation, Business Jet, and Commercial Transport). Input data for neural networks include airfoil and ice geometries, transformed into a set of parameters using a parabolic coordinate system and Fourier series expansion. Besides input features include physical parameters of flow (velocity, temperature, droplets diameter, liquid water content, time of ice accretion) and angle of attack. The novelty of this work is in that the neural network dataset includes various airfoils and the data augmentation technique being a combination of all time slices. Several artificial neural networks (ANNs), fully connected networks (FCNNs), and convolutional networks (CNNs) were trained to predict airfoil ice shapes. Two different loss functions were considered. In order to improve performance of models, batch normalization and dropout layers were used. The most accurate results of ice shape prediction were obtained using CNN and FCNN that applied batch normalization and dropout layers to output neurons of each layer.

**Keywords:** aircraft icing; airfoil; ice shape; CFD solver; simulation; dataset; loss function; neural network; CNN

## 1. Introduction

The study of the ice accretion is an important process for different aspects of people's life, management of technical transportation and energy devices such as aircraft, helicopters, Unmanned Aircraft Systems, wind turbines, and power electrical lines.

The mass of accreting ice can reach critical values and become the reason for the various technical problems or disasters.

Ice accretion on aircraft wing and helicopter rotor blades impairs aerodynamic performance thereof, which can lead to an aircraft crash. A detailed analysis of different aircraft crashes was presented in [1]. One of the recent aircraft accidents due to icing in Russia is the case of An-148 aircraft of Russian Saratov Airlines, which was flying from Moscow to Orsk (Orenburg region), which crashed on 11 February 2018 a few minutes after takeoff from Moscow's Domodedovo airport. The Interstate Aviation Committee noted that the problems could have been caused by icing of sensors of total pressure probes that measure the aircraft velocity [2].

The ice shape on the airfoil significantly depends on the icing regime or ice type. The latter could be glaze ice (low water content and temperature), rime ice (high water content

and temperature), and mixed ice (transitional from rime to glaze and vice versa). This fact is well demonstrated by NASA experiments on the NACA0012 airfoil icing cases 421–424, in which all parameters are the same, except for the temperature [3]. According to the results of the study [3], one can see that a temperature decrease causes the ice shape to change its form from a rather complex one with the presence of a pronounced horn (glaze ice mode) to a simpler one that repeats the shape of the airfoil (rime ice mode).

Currently, research projects on aircraft icing are underway in the United States, Europe, Canada, Japan, and the Russian Federation to develop the concept of new models of regional and supersonic passenger aircraft. The most famous projects are Aerion AS2, SpikeAerospace, Low Sonic Boom Configuration. Such an aircraft should be designed based on flight safety requirements, including those in difficult climatic conditions of the northern territories. The issues of studying the processes of formation of various forms of ice (rime, mixed, glaze, ridge) on different airfoils, changes in aerodynamic coefficients using experiment and mathematical modeling are relevant. The physical parameters of flow with water droplets up to 40 μm in diameter are defined in Aviation Rules, Appendix C of Part 25 [4] (Russia) or in FAA report [5] (USA).

A detailed overview of the topic with aircraft icing is given in the following scientific works [1,6–9]. There are different approaches to the study of the ice accretion process (experimental, field, mathematical modeling with special codes, Machine Learning and Neural Networks). Some research teams have used approaches with Computational Fluid Dynamics (CFD) codes and artificial neural networks. Different numerical solvers for the ice accretion process have been developed since 1980s.

Among these codes are LEWICE, CANICE, AEROMSICE-2D, PoliMICE, NSCODE-ICE, ICECREMO, FENSAP-ICE, CLORNS, IGLOO2D.

The research teams from different organizations develop their codes for ice accretion modeling: Beijing University of Aeronautics and Astronautics, China [10], Nanjing University of Aeronautics and Astronautics, China [11], Politechnico di Milano, Italy [12,13], Polytechnical Montreal University, Canada [14], University of Nottingham, UK [15], MCGill University, Canada [16], ONERA, France [17].

Over the past 10 years, a large number of articles have been published that are devoted to the application of machine learning methods, deep neural networks in the field of fluid dynamics and calculating aerohydrodynamic coefficients, and terms in the original equations reflecting conservation laws [18]. Among them are the aerodynamic drag and lift coefficients [19], the heat transfer coefficient, the induced mixing efficiency in stratified flows [20], the coefficients in engineering turbulence models [21], the Reynolds stress tensor [22], turbulent scalar flux [23], the rate of physical and chemical reactions [24].

Previously, studies for research of ice accretion process were carried out using computational fluid dynamics and neural networks to simulate the ice formation process on the airfoils. A study was carried out of two architectures of neural networks that are best suited for this task [25]:

- A multilayer neural network with a learning algorithm using the back propagation of an error;
- A network of radial-basis functions.

To study the change in the airfoil shape during icing in [26], an algorithm was used that involved two conformal mappings, namely the investigated airfoil into a parabolic coordinate system and the Prandtl transposition to transform clean airfoil into a straight line. In this case, the shape of the frozen ice was presented as a perturbation of this line. The shape of the disturbance was set by the Fourier series or using wavelet functions [26]. The number of coefficients in the expansion and their values were the objects of prediction of the neural network. The following parameters were used as input parameters for the neural network:

- Atmospheric conditions (temperature $T$ and pressure $p$);
- Flight parameters (velocity $\vec{U}$);

- Droplet diameter $d$;
- Density or water content of drops;
- Drop time or time ice accretion $t$.

The results of the NASA experiment were used for validation. CFD simulations were performed using the NASA LEWICE package [3].

When studying the coefficients of the function approximating the shape of frozen ice, it was taken into account that their number should not greatly exceed the number of input parameters, otherwise the training time of the neural network would increase and the accuracy of the predicted results would decrease. The architecture of the neural network, which is based on dense neural network and it has one hidden layer, also provided a statistical inference of the relative significance of the input parameters in training.

The work [27] considered the blade of the Sikorsky SC2110 helicopter. Using the PoliMIce library, the flows under icing conditions were calculated for different liquid water content (LWC) water conditions and median volume diameter (MVD) droplet sizes for 101 cases of airfoil. The data for the calculation were selected based on the data of the flight experiment.

Deep Neural Network, Bayesian Neural Network together with CFD calculation results in SU2 code together with Ffowcs–Williams–Hawkings acoustic analogy to calculate far-field acoustic noise spectrum were used to calculate six aerodynamic coefficients (three coefficients for force, three coefficients for torque) [27]. Based on the results of the performed comparative analysis, it can be concluded that the finite volume method is more preferable, since the basic equations reflecting the conservation laws are written in integral form. This method works effectively with structured and unstructured meshes.

The authors of [28] introduced a purely data-driven approach to find the complex pattern between different flight conditions and aircraft icing severity prediction. The Extreme Gradient Boosting Supervised Learning (XGBoost) algorithm has been applied to create a prediction framework that makes a prediction based on any set of observations.

The input flight conditions for the proposed prediction framework are liquid water content, droplet diameter and exposure time. The proposed approach was demonstrated in three cases: maximum ice thickness prediction, icing area prediction and icing severity level evaluation.

Modeling a turbulent fluid flow around aircraft, taking into account the formation of ice of various shapes in a 3D setting, is an expensive computational procedure, especially when it is necessary to perform parametric studies. Approaches based on approximation with a decrease in the dimension of the system under study referred to as the methods of dimension reduction by means of Proper Orthogonal Decomposition (POD) are a good alternative for reducing computer time.

Reduced Order Modeling (ROM)—the approach uses the results or labeled snapshots obtained under specified conditions to construct basis vectors (modes) that reliably reproduce the main features of the flow. A linear combination of these modes can be used to obtain new solutions when specifying new input parameters that differ from previously obtained solutions (labeled snapshots). To obtain a system of reduced dimensions, various methods can be used, including the POD [29,30].

In this approach, a global POD and a local POD can be used. Global POD uses all available solutions or labeled snapshots. In the case of icing simulation, different forms of ice are possible (rime ice—loose ice, glaze ice—smooth ice) when the initial parameters change. For external aerodynamics problems, different flow regimes are possible (shock waves, flow separation) when the Mach number changes for subsonic and transonic flow modes.

The local POD method handles different physical characteristics in different ways. The local approach requires dividing the solution space into separate subdomains, each of which ideally contains solutions characterized by similar or sufficiently close physical structures. With solutions from each solution cluster, POD's linear approach allows for a generic solution using multiple modes. In several papers, the k-means algorithm, one of the machine learning methods, has been used to develop the local POD method.

As a rule, a software implementation of libraries for machine learning is done in Python programming language, using numerical libraries and the open frameworks such as Keras, scikit-learn, PyTorch, TensorFlow [31].

A significant disadvantage of many solvers for modeling icing is high computational costs: it takes several hours to several days to calculate a two-dimensional airfoil involving supercomputer resources.

The main goal of the current research is to build a model based on artificial intelligence (neural networks) to accelerate numerical calculations obtained using CFD solvers. The combination of two approaches: machine learning and numerical simulation, will speed up the prediction of the icing shape and significantly reduce computational costs.

The main part of this paper has the following structure. Section 2 contains a description of the mathematical model for ice accretion simulation. Section 3 describes the Definition of the problem for 2D airfoil. Section 4 describes Materials and Methods. Section 5 contains the results of simulations with neural networks. Section 6 contains Discussion. Section 7 concludes the paper.

## 2. Mathematical Model for Ice Accretion Simulation

The mathematical models for ice accretion may include Euler–Euler and Euler–Lagrangian models, a hybrid method with panel and integral Boundary Layer methods. To calculate the ice shape iceFoam CFD solver is used which was developed in ISP RAS, and which is based on the Euler–Lagrangian method and the SWIM model for ice and fluid film simulation [32].

The iceFoam solver is being developed on the basis of OpenFOAM package [33]. To describe the gas-droplet medium, the Euler–Lagrange model is used, which is based on a system of continuity, momentum, and energy equations, and the finite volume method for solving the government equations [34].

The iceFoam solver uses the PIMPLE algorithm to solve the velocity and pressure equations. When the gas-droplet flow interacts with the irregularities and the roughness of the solid surface of the body, an ice film and a liquid water film may appear and grow.

This approach requires a separate unstructured mesh for the thin film area. As for the particles, the right-hand side of the equations for mass and energy contains source terms that characterize the processes of particles melting, splashing, convective heat transfer. The ice accretion leads to a change in the initial shape of the body. The boundary of the body moves in space along the normal.

At the same time, it is necessary to ensure the simultaneous movement of borders for two different grids in the calculation program and recalculation of the position of grid nodes using the solution of the Laplace equation. To characterize the ability of the curved surface of the body to capture liquid drops, the water collection efficiency coefficient $\beta$ is used, and to describe thermal processes, the coefficient of heat transfer is applied.

The medium under consideration is a non-reacting equilibrium mixture of gases with a total temperature $T$, density $\rho$, and partial pressures $p_i$ for various components of the mixture.

In the framework of the selected mixture approximation, it is assumed that the mass, momentum and energy of the entire flow is transferred by the mass-averaged velocity $\vec{U}$ and the mass fraction of the mixture components of the flow incident on the body under study does not change with time.

The mutual motion of the mixture components is taken into account in the diffusion approximation. The effect of the dispersed phase on the continuous one is introduced as additional terms in the equations.

The mass conservation equation for the mixture:

$$\frac{\partial \rho}{\partial t} + \nabla \cdot (\vec{U}\rho) = \dot{\rho}_v, \tag{1}$$

The momentum balance equation for the mixture:

$$\frac{\partial \rho \vec{U}}{\partial t} + \nabla \cdot \left( \vec{U} \rho \vec{U} \right) + \sum_i \rho_i^0 \vec{W}_i \vec{W}_i = \dot{\rho}_v \vec{U}_v + \nabla \cdot \hat{\sigma} - \nabla p. \tag{2}$$

The energy balance equation with specific enthalpy for the mixture:

$$\frac{\partial \rho h}{\partial t} + \nabla \cdot \left( \vec{U} \rho h \right) + \frac{\partial \rho K}{\partial t} + \nabla \cdot \left( \vec{U} \rho K \right) + \sum_i \nabla \cdot \vec{W}_i \rho_i^0 e_i - \frac{\partial p}{\partial t} =$$

$$= -\nabla \cdot (\hat{\sigma} \cdot \vec{U}) - \nabla \cdot \vec{q} + \dot{\rho}_v e_v, \tag{3}$$

where $\rho$ is the density of the mixture, $\rho_i^0$ is the density of the $i$-th component, $\dot{\rho}_v$ is a source term describing the mass transfer between the gas and droplet phases, $\vec{W}_i$ is the relative speed, $\dot{\rho}_v \vec{U}_v$ is the exchange of momentum between the environment and drops of particles, $p$ is the ambient pressure, $\hat{\sigma} = \mu \left( \nabla \vec{U} + (\nabla \vec{U})^T \right) - \frac{2}{3} \mu I \nabla \cdot \vec{U}$ is the viscous stress tensor; $\mu$ is the coefficient of viscosity of the mixture, $I$) is the identity tensor, $e$ is the internal energy of the mixture, $h$ is the specific enthalpy of the mixture, $K$ is the turbulence kinetic energy, $\dot{\rho}_v e_v$ is the exchange of energy between the environment and drops the particles.

The heat flux vector is calculated following Fourier's law $\vec{q} = -\lambda \nabla T$, where $\lambda$—the thermal conductivity of the mixture:

$$C_p = \left( \frac{\partial h}{\partial T} \right)_p, \quad \nabla T = \frac{\nabla h}{C_p}.$$

The specific enthalpy of a mixture is the weighted sum of the enthalpies of its components: $h = \sum_i Y_i h_i$, where $Y_i = \dfrac{\rho_i^0}{\rho}$ is the mass fraction of the $i$-th component.

The mass balance equation of $i$-th component:

$$\frac{\partial \rho Y_i}{\partial t} + \nabla \cdot \left( \vec{U} \rho Y_i \right) + \nabla \cdot \vec{W}_i \rho_i^0 = 0. \tag{4}$$

The closing ratio for mass fractions of the mixture: $\sum_i Y_i = 1$.

The average mass velocity $\vec{U}$ and relative velocities $\vec{W}_i$ are entered so that:

$$\vec{U} = \frac{\sum_i \rho_i^0 \vec{U}_i}{\rho}, \quad \vec{W}_i = \vec{U} - \vec{U}_i, \quad \sum_i Y_i \vec{W}_i = 0.$$

To calculate the relative velocities of the gas components, the diffusion approximation is used:

$$\rho_i^0 \vec{W}_i = -D_i \nabla \rho_i^0,$$

where $D_i$ is the diffusion coefficient of the component: $D_i = \dfrac{\nu_i}{\text{Sc}}$ (the values of the number Sc are calculated from the tables of medium properties depending on the temperature and composition of the mixture).

All components of the gaseous mixture are a perfect gas with a constant molar mass:

$$p = \rho R \sum_i \frac{Y_i}{M_i} T,$$

where $R$ is the gas constant, $M_i$ is the molar mass of the $i$-th component.

The OpenFOAM particle cloud model sprayCloud is used as the base model. A cloud of spherical droplets–particles is determined by the position of its center of mass $\vec{x}_p$, the diameter of the incoming drops $D_p$, the speed of the drops $\vec{U}_p$, and the density of the substance $\rho_p$.

Then the mass of single particle:

$$m_p = \frac{1}{6}\rho_p \pi D_p^3.$$

The particles with similar parameters are represented by a cloud, since simulating all real droplet particles separately is expensive for computing resources. The clouds of particles do not interact with each other.

The trajectory of the particle cloud is determined by integrating the kinematics equation:

$$\frac{d\vec{x}_p}{dt} = \vec{U}_p. \tag{5}$$

The particle velocity is determined from the solution of the force balance equation. The force acting on a particle is the sum of all the forces acting. Examples of such forces are the environmental drag force, gravity, buoyancy, and pressure force:

$$m_p \frac{d\vec{U}_p}{dt} = \sum \vec{F}_i = \vec{F}_D + \vec{F}_G = \frac{3}{4}\frac{m_p \mu C_D \mathrm{Re}_p}{\rho_p D_p^2}(\vec{U} - \vec{U}_p) + m_p \vec{g}(1 - \frac{\rho}{\rho_p}), \tag{6}$$

where $\vec{F}_D$ is the pressure force and $\vec{F}_G$ is the gravity force.

The $C_D \mathrm{Re}_p$ complex is calculated depending on the selected model for calculating the drop drag coefficient using a function that depends on the Reynolds number of the particle $\mathrm{Re}_p$.

There are available models: Putnam; Habashi; Prikhodko; Gent; Ochkov; Schiller–Neumann.

The Reynolds number for particle: $\mathrm{Re}_p = \frac{\rho|\vec{U} - \vec{U}_p|D_p}{\mu}$.

The drop model also includes a drop mass balance equation:

$$\frac{dm_p}{dt} = \dot{m}_p = 0, \tag{7}$$

and the heat balance equation for the drop:

$$m_p C_{\mathrm{pp}} \frac{dT_p}{dt} = Q_T \tag{8}$$

where $C_{\mathrm{pp}}$ is the specific heat capacity at constant drop pressure; $T_p$ is the average volume temperature of the drop.

As a result of convective interaction with the main flow, the droplet jets take or give away part of the internal energy of the gas flow. Heat flow from the environment:

$$Q_T = \mathrm{htc}_p \times S_p \times (T_{\mathrm{sp}} - T), \tag{9}$$

where $S_p = \pi D_p^2$ is the surface area of the drop; $T$ is the ambient temperature.

The surface temperature drops $T_{\mathrm{sp}}$:

$$T_{\mathrm{sp}} = \frac{2}{3}T_p + \frac{1}{3}T. \tag{10}$$

Heat transfer coefficient htc:

$$\mathrm{htc}_p = \frac{\mathrm{Nu} \times \lambda}{D_p}, \tag{11}$$

where $\lambda$ is the coefficient of thermal conductivity of the environment.

There are four models to choose from for calculation of heat transfer between the drop and the surrounding gas: Clift; Feng; Ranz–Marshall and Whitaker. The most widely used

model is the Ranz–Marshall model, which uses coefficients to calculate the Nusselt number Nu for a spherical drop:

$$\mathrm{Nu} = 2 + 0.6\sqrt{\mathrm{Re}_p} \times \sqrt[3]{\mathrm{Pr}}. \tag{12}$$

The Prandtl number of the gas $\mathrm{Pr} = \dfrac{c_p \mu}{\lambda}$ where $c_p$ is the specific heat capacity of the environment at constant pressure; $\mu$ is the dynamic viscosity of the surrounding gas.

To model film layers in OpenFOAM, one needs to select a special outer area of the grid from the aerodynamic region [35].

All parameters of the thin film are calculated in this selected area of the grid. This grid area can be created using two OpenFOAM utilities. First, the topoSet utility is used to extract all the cell faces of a section of the airfoil from the existing aerodynamic grid, and the extracted set of cell faces is used to extrude a new area of the grid using the extrudeToRegionMesh utility. The extrudeToRegionMeshDict dictionary sets parameters such as the set of cell faces to use, the number of layers, and the extrusion thickness.

To model the film layer, the so-called thin-film approximation is used, which means that the velocity normal to the grid on the wall is assumed to be zero. In addition, the tangential near-wall diffusion is considered insignificant compared to the normal near-wall diffusion.

### 2.1. The SWIM Model for Liquid Film

Various models are used to simulate thermodynamic processes of icing on the surface. Among them are the following models: the Messinger model [36], Iterative Messinger Model [37], and Myers model [38]. More details about thermodynamic models can be found in this review [39].

In addition, note that film transport models were developed in NASA LEWICE2D, 3D code [40,41], in IGLOO2D ONERA code [42,43], in works from Iowa State University [44,45], in PoliMICE code Politecnico di Milano [12,46].

The SWIM is a Partial Differential Equation (PDE) developed for calculating the ice accretion process in its original form. It is represented by the following equations. The SWIM model is given in [47,48].

The flow in the wall film is calculated using the mass conservation equation:

$$\frac{\partial \rho_w h_w}{\partial t} + \nabla \cdot (\rho_w h_w \vec{u}) = S_{imp} - S_{ice} \tag{13}$$

where $t$ is time, $\rho_w$ is the water density, $h_w$ is the thickness of the water film layer, $\vec{u}$ is the water film velocity, $S_{imp}$ is the mass added to the film layer due to particle collisions, and $S_{ice}$ is the mass change due to water solidification into ice.

The momentum balance equation in the original SWIM model:

$$\vec{u} = \frac{h_w}{2\mu_w} \vec{\tau}_{wall} \tag{14}$$

where $\mu_w$ is the dynamic viscosity of water, $\tau_{wall}$ is the air wall shear stress.

The energy balance equation:

$$\frac{\partial \rho_w h_w H}{\partial t} + \nabla \cdot (\rho_w h_w \vec{u} H) = S_{imp} \frac{\vec{U}_{imp}^2}{2} + Q_c - Q_{wall} +$$
$$+ S_{imp} C_w (T_{imp} - T) - S_{ice} C_{ice} \left(T - T_{ref}\right) + S_{ice} L_f \tag{15}$$

where $H$ is the enthalpy of water, $U_{imp}$ is the impact droplet velocity, $Q_c$ is the convective heat transfer, $Q_{wall}$ is the heat transfer to the airfoil surface, $C_w$ is the specific heat of water, $T_{imp}$ is the impinging droplets temperature, $T$ is the water film temperature, $C_{ice}$ is the

specific heat of ice, $T_{ref}$ is the temperature of the triple point of water, $L_f$ is the fusion latent heat of ice.

The ice thickness $h_{ice}$ is determined by the formula:

$$\rho_{ice}\frac{\partial h_{ice}}{\partial t} = S_{ice}, \tag{16}$$

where $\rho_{ice}$ is the density of ice.

The system of equations has 5 unknowns $h_w, \vec{u}, T, S_{ice}, h_{ice}$ and cannot be solved directly. The main assumption of the SWIM model is that the film temperature is equal to the temperature of the triple point of water ($T = T_{ref}$). In this case, one can calculate all the unknowns, including the thickness of the water film and the thickness of the ice. In particular, the energy equation, assuming that the enthalpy is a function of temperature only $H_f = H\left(T_{ref}\right)$, takes the form:

$$S_{ice} = \frac{S_{imp}H_f - S_{imp}\frac{\vec{U}_{imp}^2}{2} - Q_c + Q_{wall} - S_{imp}C_w\left(T_{imp} - T_{ref}\right)}{H_f + L_f} \tag{17}$$

This paper introduces a modification of the SWIM model, and the momentum balance equation has the form:

$$\frac{\partial \rho_w h_w \vec{u}}{\partial t} + \nabla \cdot (\rho_w h_w \vec{u}\vec{u}) = -h_w\nabla p + \vec{S}_{\rho\delta\vec{U}} + \vec{\tau}, \tag{18}$$

where $p$ is the pressure, $\vec{S}_{\rho\delta\vec{U}}$ is the contribution from the falling drops, and $\tau$ is the stress from the forces acting on the film.

The heat transfer coefficient (htc) is a very important parameter in the simulation of ice accretion for airfoils and wings. The predicted ice shape can be quite different from the actual scenario if the predicted htc is inaccurate.

In our model, the heat transfer coefficient htc is calculated using an empirical formula, for which the spatial length scale $L$ must be specified. In the presence of turbulence, it is assumed that the effective values of the corresponding quantities are used. In what follows, the generally accepted variable notation is used. The local Reynolds and Prandtl numbers are calculated for the boundary cells.

$$\text{Re} = \frac{\rho|\vec{U}|L}{\mu}, \tag{19}$$

$$\text{Pr} = \frac{\mu C_p}{\lambda}. \tag{20}$$

$$\text{htc} = \begin{cases} 0.664\sqrt{\text{Re}}\sqrt[3]{\text{Pr}} \times \lambda/L, & \text{Re} < 5 \times 10^5; \\ 0.037\sqrt{\text{Re}}^{-0.8}\sqrt[3]{\text{Pr}} \times \lambda/L, & \text{Re} \geq 5 \times 10^5. \end{cases} \tag{21}$$

The mathematical model is complemented by Reynolds-averaged Navier–Stokes (RANS) turbulence models $k$-$\varepsilon$, $k$-$\omega$ SST, Spalart–Allmaras [49].

When modeling icing, the process of convective heat transfer has a great influence on the ice shape, on the presence of roughness and is the key to solving the process of convective heat transfer.

There are several works devoted to heat transfer coefficient determination [41,50–54]. However, because in the SWIM model the temperature of the water film and the temperature of the ice are constant and equal to the temperature of the triple point, it became possible to use traditional formulas for the transfer coefficient. The results obtained using this model are in good agreement with the experiment.

Currently, most icing models use the boundary layer cumulative functions proposed in the LEWICE model, which take surface roughness and velocity variation into account to solve the heat transfer coefficient.

The roughness influence problem for ice accretion simulation in the case of Appendix C of Part 25 of Title 14 in the U.S. Code of Federal Regulations (CFR-25) [5] is relevant and was studied in several works [55–61].

The effect of ice roughness was taken into account through the roughness parameter in the RANS SA [59] and $k$-$\omega$ SST [60,61] turbulence models through the near-wall function model [59–62].

Two models are considered in the literature for determining the value of Ks (the equivalent sand-grain roughness height):

1. Constant roughness models: Equivalent sand-grain models [55–58,63,64];
2. Non-uniform roughness models [65–68].

To calculate the value of the roughness height Ks, we used an approach based on an empirical model for taking into account the influence of the airfoil chord size $C$ from [55] and NASA model implemented in LEWICE3D calculation code [56–58].

For non-uniform roughness, the root mean square roughness height is calculated for each codebook vector using the theoretical Self-Organizing Map—SOM model [67]. The iceFoam solver implements the SA and k-omega SST models, taking into account the "Equivalent sand-grain" model. Currently, work is underway within the framework of the European project ICE-GENESIS. A separate scientific group (ONERA, CIRA, TUDA, TUBS, PoliMI) is developing roughness models for CFR-25 Appendix C [69].

### 2.2. The Features of Implementing iceFoam Solver in OpenFOAM Package

### 2.2.1. Implementing a Dynamic Grid

As part of the OpenFOAM package, it is necessary to implement the movement of grid nodes in two regions, i.e., in the area of the external gas-droplet flow and inside the film. The actual film grid in OpenFOAM approach is only one cell thick. The film cell size by thickness does not make physical sense and does not change when the calculated film thickness is changed. This is explained by the concept of OpenFOAM package, which always solves 3D equations, even when they are described in 1D or 2D space in the original formulation. However, one should move the grid nodes according to the changing ice boundaries.

One of the proven algebraic methods for moving the grid is the bisector method.

Since the nodes to change are located on the border between the two regions, it is necessary to rebuild the grid in the gas-droplet flow area. To do this, the standard OpenFOAM procedure based on solving the Laplace equation was used. As a boundary condition for it, the displacement of the ice boundary nodes is set. The solution gives the offset of all other nodes in the gas-droplet flow area.

### 2.2.2. Different Versions of the iceFoam Solver

A total of 3 versions of the iceFoam solver have been developed based on the Euler approach for the gas phase, the Lagrangian approach for modeling water droplets and a model of a water film on the surface of an airfoil over an ice layer. All three solvers use a water film and ice layer model that is linked to a shallow water model.

The first version of the iceFoam solver is intended for initial estimation of icing spots of an arbitrary 3D airfoil. This solver fully uses all the capabilities of Open-MPI technology for efficient parallel computations using the open-source package OpenFOAM.

The main limitation of this solver is the assumption that the thickness of the ice layer is sufficiently small and the change in the airfoil surface during icing can be neglected. The airfoil with ice and the ice-free airfoil are assumed to be the same. The developed extrudeToFilmCellDist utility is used to decompose the spatial region of the film. This utility makes it possible to ensure uniformity of domain decomposition in the gas domain and in the adjacent film. This version has no restrictions on the number of computational cores.

The second version of the iceDyMFoam2 solver is designed to simulate icing with both rime ice and glaze ice, for which the ice surface often takes complex and bizarre shapes. In this version, the solver quite effectively allows consideration of the effect of changes in the airfoil surface during icing on the icing process itself. This solver requires the use of a structured mesh near the airfoil surface and the presence of only one spatial domain for the film and the adjacent Euler mesh. Computational experience has shown that this requirement limits the maximum number of computational cores. With a further increase in the number of computational cores used, the efficiency of parallel ones grows weakly or even decreases.

To simplify the decomposition of the spatial gas domain, the addTwoLayersTo0 utility has been developed. This utility allows us to set the computational domain on the zero processor for the film and for two adjacent layers of the Eulerian grid of the gas phase of the flow.

The third version of the iceDyMFoam3 solver also allows us to take into account the effect of changes in the airfoil surface during icing on the icing process itself. Like the first version, this solver fully uses all the capabilities of the Open-MPI parallel library technology to perform efficient parallel computations. Parallel implementation of the ice boundary required the use of low-level functions of the OpenFOAM package. In this case, there will necessarily be nodes that fall on the border of several domains. In the case of using the dynamic grid approach, it was necessary to collect information about nodes that are stored in several domains or on different computational cores. For such nodes, after local calculations within one domain, an operation was performed to determine the new position of the node.

Next, averaging over all domains participating in the calculations of this node was performed. The averaged values of the position of nodes, which were recorded in more than one domain, were distributed back across all the necessary computational cores. Thus, the coincidence of moving nodes on different domains was ensured. For the case of icing with glaze ice, the third version of the iceDyMFoam3 solver is still inferior in its capabilities to the second version of the solver in terms of reliability. The extrudeToFilmCellDist utility is also used to decompose the spatial region of the film.

### 3. Definition of the Problem for the 2D Airfoil

In this paper, the initial boundary value problem for the case of flow around 2D airfoils is formulated.

The numerical simulations for 2D airfoils (NACA0012, Business Jet, Commercial Transport, General Aviation) with different angles of attack were performed using domain with size $x = [-1; 2]$ m, $z = [-1; 1]$ m.

The computational domain had a form of a hemisphere near the inlet boundary and a rectangle shape at the other borders. The hexahedral grid included several blocks with clusters of mesh refinement in areas near the airfoil surface and the wake.

To determine the optimal mesh parameters in the context of case 421 of the NACA 0012 airfoil, see Table A1, calculations were carried out for three mesh divisions. For the first variant, there were 4000, 30, and 15 cells for the entire area, the film area, and the ice growth zone, for the second, 16,000, 120, and 30, and the third, 20,800, 180, and 60 cells, respectively. Figure 1 shows the mesh of second option.

A comparison of the calculation results for overgrown ice for the three options is shown in Figure 1. The figure shows that an increase in the number of nodes has a good effect on the results of calculating the position of the edges of the ice cover (especially the upper horn). To achieve the necessary compromise between calculation accuracy and calculation time, we finally chose the option with 30 calculation cells (second option, Figure 1) for the leading edge for all airfoil options listed in Table A1.

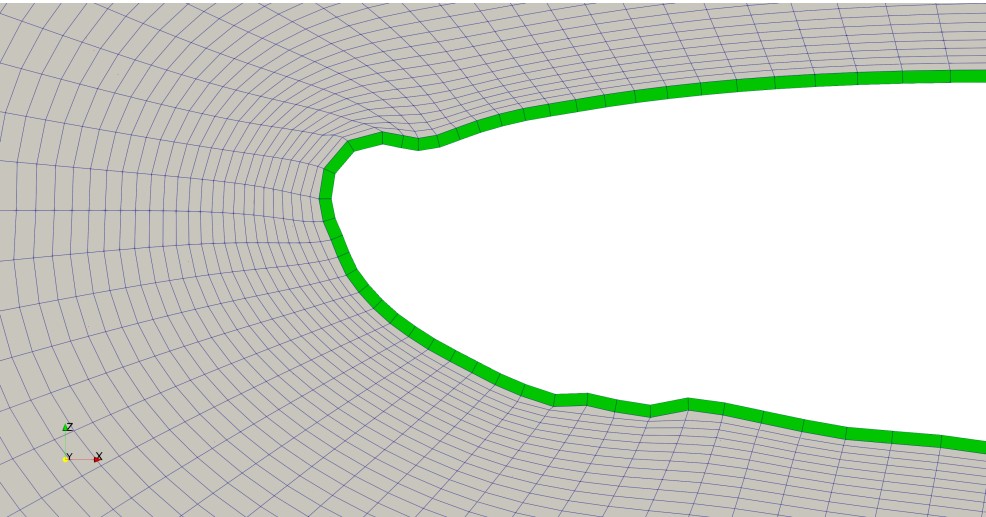

**Figure 1.** Mesh for simulation for the 421 RUN case.

The numerical grid was built using the blockMesh, extrudeMesh and extrudeToRegionMesh OpenFOAM utilities [33]. The y+ average value for coarse mesh for the first near-wall cells was 5.

The boundary conditions for velocity were the following: for "inlet" it is Dirichlet boundary condition, for "outlet" it is Neumann boundary condition, for "airfoil" it is no-slip boundary condition. For pressure the boundary conditions for inlet and outlet were calculated.

To approximate the terms in the time the Euler scheme is used, to approximate the inviscid terms the first order upwind scheme is used and to approximate the viscous terms the second order linear corrected scheme is used.

After discretization of the terms in the basic equations, the linear algebraic equations were solved numerically. The smoothSolver with smoother symGaussSeidel was used for calculation of velocity, the GAMG method with smoother GaussZeidel was used for calculation of pressure, the PBiCGStab method with preconditioner DILU was used for calculation of enthalpy.

The initial position of spherical particles was set at the entrance to the computational domain, as well as the frequency with which the particles were introduced into the computational domain and the total mass of the particles. The temperature of the liquid particles was set equal to the ambient temperature. Thus, it was possible to calculate the LWC. The Reynolds equations and the SST $k$-$\omega$ turbulence model with wall functions were used to describe the gas-droplet medium. The time step was chosen according to the local Courant number. The Courant number ($Co$) varied in the range from 0.3 to 0.5 far from the airfoil, to about 10 in the boundary layer near the airfoil surface. A large value of the local Courant number ($Co = 10$) does not lead to loss of stability of the numerical solution, since the equations in the boundary layer zone are not hyperbolic, and the solution method is semi-implicit.

The input parameters of the flow velocity correspond for Run case 421 (See Appendix A).

Figure 1 shows the distribution of the thickness of the ice film over the surface and velocity fields on the dynamic mesh. Figure 2 shows the domain decomposition for parallel simulations on the high-performance cluster. Figure 3 presents the results of calculations of flow around NACA0012 for Run case 421 and comparison with experiments of NASA [70,71]. Figure 4 presents the results of calculations of flow around NACA0012 for Run case 425 and comparison with the experiment of NASA [70,71]. The value of length scale $L$ for the calculation of htc in case Run 421 was set to 0.4.

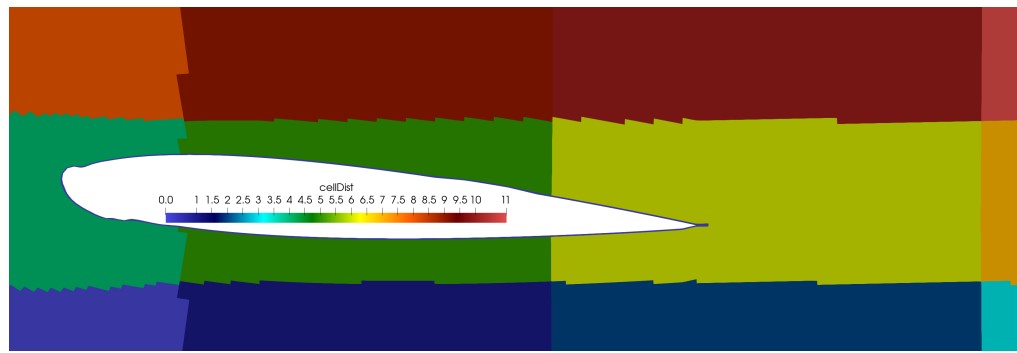

**Figure 2.** Results of domain decomposition for the 421 RUN case.

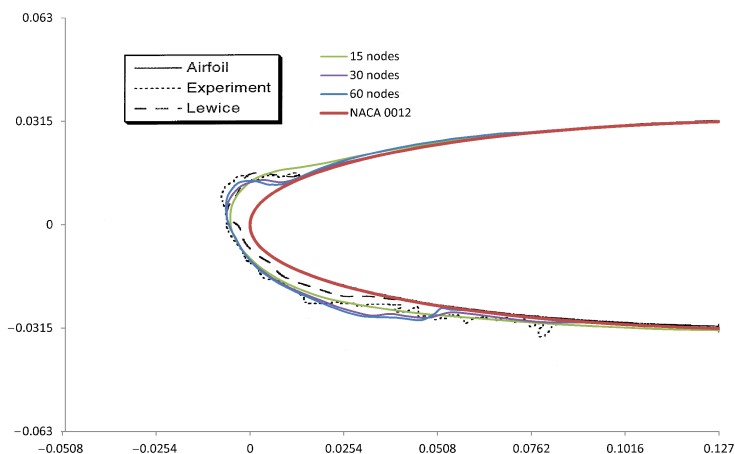

**Figure 3.** Results of ice simulation for the 421 RUN case for different meshes.

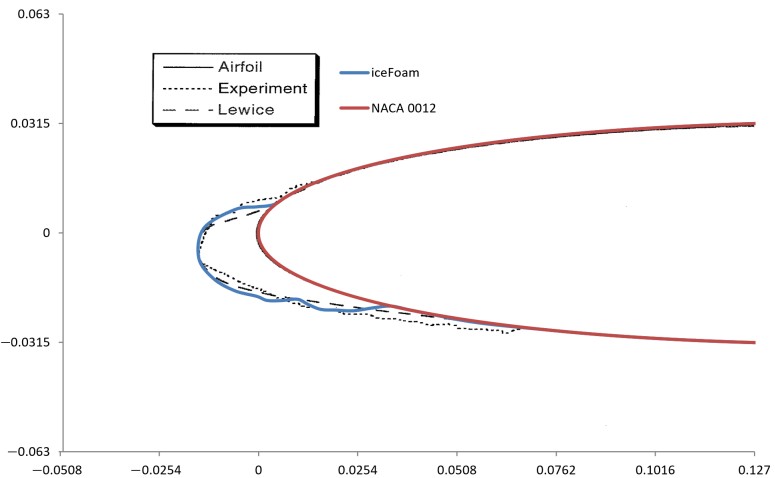

**Figure 4.** Results of ice simulation for the 425 RUN case.

The dotted line shows the results of the NASA experiment, the dashed line denotes the results obtained using the NASA LEWICE code, the color vector denotes the results obtained using the iceDyMFoam3 solver. The time step was about $2 \times 10^{-4}$ s, and the number of particle packets was about 4000. The 6 min of model icing time calculation takes about 60 h on a server with 12 computational cores.

The number of particles in the pack was equal to 140,000. All the data for cases are mentioned in Table A1, Appendix A.

When the temperature changes, various forms of ice may form on airfoil as rime, glaze or mixed ices. One of the important requirements for the solver is the ability to run it in parallel mode. This allows us to significantly reduce the time spent on the simulation.

The numerical calculations of icing require significant resources from the high-performance cluster. The use of neural networks can significantly reduce the use of the expensive computer's time. The issues of choosing the optimal architecture of a neural network, the choice of input parameters and the required metrics are topical.

## 4. Materials and Methods

### 4.1. Generation of Datasets

The ice shape dataset was obtained for 4 different 2D airfoils (NACA0012, Business Jet, Commercial Transport, General Aviation). The geometry for airfoils is described in [70,71], the description of considered cases can be seen in Appendix A. The final dataset includes the ice thickness which was represented by two coordinates $x$, $y$, time steps, and 6 physical parameters which determined the condition of ice accretion.

The shape of the ice on the airfoil can be approximate using a mathematical function. Further research was conducted related to the analysis of this function and its representation using the Fourier transform.

### 4.2. Analysis of Shape Function

Parametrization of the airfoil shape and the ice formed on it is a difficult problem. The shapes of different airfoils are complex curves that cannot be parametrized with a single set of coefficients. On the other hand, the grid representation of geometry in the form of a set of points with coordinates $(x, y)$ also cannot be used as parameters, since, firstly, this set of parameters strongly depends on the choice of the grid and secondly the number of such parameters will grow rapidly with decreasing size of the grid cell. To solve this problem, the approach implemented in [25] was used. The main idea of this approach is to use the transformation of an airfoil shape into a function, which can then be expanded into a Fourier series. The Fourier coefficients will unambiguously determine the specified airfoil.

Let us briefly describe the process of transition from an airfoil to a set of Fourier coefficients.

As a result of the calculation, there is a set of ordered points that determines the state of the airfoil depending on the given moment in time $t_i$: $(x_j(t_i), y_j(t_i)), j = \overline{1, N_{grid}}, i = \overline{1, M}$, where $N_{grid}$ is the number of grid elements that define the airfoil geometry, and $M$ is the number of time slices.

To transform an airfoil curve into a function, it is necessary to use a parabolic coordinate system. On the other hand, for the points of the investigated airfoils to fit better on the ground, it is necessary to scale them by dividing them by the leading edge radius ($r_l$):

$$x_i' = x_i/r_l - 0.5, \quad y_i' = y_i/r_l. \tag{22}$$

The $x$-axis shift by 0.5 to the right is necessary so that the leading edge of the airfoil lies inside the parabola.

The $r_l$ value is determined by the airfoil, however for a uniform conversion of all airfoils, the average value $r_l = 0.03c$ was used, where $c$ is the chord length. All airfoils (except NACA0012) have a chord length of approximately 1 m, so the radius is $r_l = 0.03$ m.

Then the transformation into the parabolic coordinate system $(\xi, \eta)$ is performed:

$$\xi_i = \text{sgn}(y_i')\sqrt{x_i' + \sqrt{x_i' + y_i'}}, \quad \eta_i = \sqrt{-x_i' + \sqrt{x_i' + y_i'}}. \tag{23}$$

Thus, representing the airfoil as a function expressed by a set of points $\eta = f(\xi)$, one can write out formulas for determining $N$ coefficients of the Fourier series expansion on some segments $[\xi_a, \xi_b]$, the length of which is $L_{ab} = \xi_b - \xi_a$:

$$a_k = \frac{2}{L_{ab}} \int_{\xi_a}^{\xi_b} f(\xi) \cos\left(\frac{2\pi k\xi}{L_{ab}}\right) d\xi, \quad b_k = \frac{2}{L_{ab}} \int_{\xi_a}^{\xi_b} f(\xi) \sin\left(\frac{2\pi k\xi}{L_{ab}}\right) d\xi, \quad k = \overline{0, N}. \quad (24)$$

Note that these integrals are calculated numerically, for example, using the Simpson formula, and using the already given partition $(\xi_i, \eta_i)$. This partition can be supplemented with intermediate points using interpolation to calculate the Fourier coefficients with large numbers since their integrands oscillate strongly.

Then, summing up the corresponding Fourier series, one can obtain an approximation of the function $f(\xi)$:

$$\tilde{f}(\xi) = \frac{a_0}{2} + \sum_{k=1}^{N} \left( a_k \cos\left(\frac{2\pi k\xi}{L_{ab}}\right) + b_k \sin\left(\frac{2\pi k\xi}{L_{ab}}\right) \right). \quad (25)$$

It is important to note that the resulting expression (25) is a function that allows restoration of the shape of the airfoil at any point on the segment $[\xi_a, \xi_b]$.

To predict the thickness of accumulated ice, it is not necessary to consider the entire airfoil. For the studied airfoils, the change in the function representing the displacement of the geometry due to the ice accretion is localized on the segment $[-4; 4]$ therefore next it is assumed that

$$\xi_a = -4, \quad \xi_b = 4, \quad L_{ab} = 8.$$

### 4.3. Augmentation of Data

One of the novelties of this work is training a neural network on various airfoils. Due to this fact it is necessary to include in the set of training parameters the geometry of the original airfoil, which has the form of a set of Fourier coefficients $\{a_k^{in}, b_k^{in}\}$. It is not necessary to look for the result in the form of Fourier coefficients required to restore the geometry. It is enough to obtain the coefficients representing the geometry change $\{\tilde{a}_k^{out}, \tilde{b}_k^{out}\}$, then the resulting airfoil shape can be restored by adding the input and output coefficients:

$$\{a_k^{res}, b_k^{res}\} = \{a_k^{in}, b_k^{in}\} + \{\tilde{a}_k^{out}, \tilde{b}_k^{out}\}, \quad k = \overline{1, N}. \quad (26)$$

Another novelty is the use of all time slices as input to learning. The main goal of forecasting using machine learning is to obtain the final ice build-up on a clean airfoil under given external conditions at time $t$. Since the source of data in this study is a numerical calculation, instead of a clean airfoil, one can consider every intermediate time moment as the initial state. This approach allows us to significantly increase the amount of data for training a neural network.

Suppose that some calculations case contain $M$ moments in time: $t_0 = 0, t_1, t_2, \dots, t_{M-1}$. Let us find the total number of training examples, which will be obtained by augmentation for this case, i.e., let us count the number of possible pairs defined for two times $\{t_i, t_j\}$, where $t_i < t_j$. For time $t_0$, one can choose $M - 1$ final icing states, i.e., transitions $t_0 \to t_1, t_0 \to t_2, \dots t_0 \to tM - 1$. Similarly, with $t_1$ there are $M - 2$ pairs: $t_1 \to t_2, t_1 \to t_3, \dots t_1 \to t_{M-1}$, for $t_3$ $M - 3$ pairs, and so on. For a moment in time $t_{M-2}$ only one transition $t_{M-2} \to t_{M-1}$, is possible, i.e., one pair. Then the number of pairs for each point in time forms an arithmetic progression with a step of 1. The sum of this progression is the total number of training pairs $\{t_i, t_j\}$, the formula for which is a function of $M$ and has the form

$$S(M) = \frac{M(M-1)}{2}. \quad (27)$$

### 4.4. Features Definition

Besides the main icing parameters: velocity $\vec{U}$, temperature $T$, liquid water content $LWC$, the droplets diameter $MVD$, time of icing accretion $t$ and angle of attack $AOA$, the airfoil shape are suggested for use as the input feature. This allows us to account for the

airfoil form changing and to predict ice shape for different airfoils. According to Section 4.2, the airfoil shape is represented by $2N + 1$ coefficients. To find a better value for $N$ many numerical experiments were conducted. It was shown that, on the one hand, if $N \leq 10$, the function of ice shape will have oscillations due to lack of high-frequency terms in (25). On the other hand, the contribution of high-frequency terms is significantly decreasing with increasing $N$ due to the Fourier coefficients properties. In this work, the $N = 20$ as optimal value for models performance. Therefore the overall number of the input parameters is 47.

The output parameters for prediction are the $2N + 1$ Fourier coefficients, which represent the form of the airfoil with accumulated ice.

### 4.5. Metric Definitions

There are two different metrics for error estimation. The first one is the Mean Squared Error (MSE)

$$err_{\text{MSE}} = \frac{1}{N_f} \sum_{k=1}^{N_f} \left( c_k^p - c_k^t \right)^2, \quad N_f = 2N + 1, \tag{28}$$

where $c_k$ denotes both sine $b_k$ and cosine $a_k$ Fourier coefficients from (24). Hereinafter, upper indexes $p$ and $t$ mean *prediction* and *target*, respectively. An $err_{\text{MSE}}$ is applied directly to the Fourier coefficients, unlike the error $e\tilde{r}r$ which is usually used to estimate the ice shape prediction error [25,26] and applied to the function deduced from the Fourier series (25):

$$e\tilde{r}r = \frac{\int_{\xi_a}^{\tilde{\xi}_b} |f^p - f^t| dx}{\int_{\xi_a}^{\tilde{\xi}_b} f^t dx}. \tag{29}$$

In this paper, the intersection over union (IoU) error $err_{\text{IoU}}$ is suggested, which is the ratio of the ice shape difference area and the total area of the predicted and target ice shapes:

$$err_{\text{IoU}} = \frac{\int_{\zeta_a}^{\tilde{\xi}_b} |f^p - f^t| dx}{2 \int_{\zeta_a}^{\tilde{\xi}_b} (|f^p| + |f^t|) dx - \int_{\zeta_a}^{\tilde{\xi}_b} |f^p - f^t| dx} \tag{30}$$

### 4.6. Neural Networks

The development of ice on airfoils using models with two different architectures of neural networks, FCNN and CNN, is studied. The choice of such architectures was based on the classical concepts of neural networks. The training procedure is the optimization of the coefficients of connections between neurons. The training algorithm is various modifications of the gradient descent are usually used to train models. According to the division of the complete dataset into training, validation, and test sets, training was carried out in batches, each batch containing 32 examples from the training set. Splitting into batches is needed to stabilize training.

A fully connected neural network (FCNN) is the simplest concept of architecture. Each neuron connects with each neuron in a subsequent layer, see Figure 5. Input and output layers are determined by the problem. The number of hidden layers and neurons on them is not defined and can be arbitrary. FCNN has a disadvantage in the curse of dimension, even a small increase in the number of layers and neurons leads to a significant increase in learning parameters.

In addition to the fully connected neural network, Convolutional Neural Networks were used. CNN is very popular today to use in image analysis for unsolved problems in medicine and in satellite imagery analysis for rapid response to natural hazards (Floods, Landslide, Water Pollution) on Earth [72–74].

Convolutional layers have significantly fewer training parameters, which gives finer possibilities for determining the complexity of the model. Filters are considered to generalize spatial structures, Figure 6. The use of convolutional neural networks in this task is due to the hypothesis of the influence of ice accretion on each other during their formation.

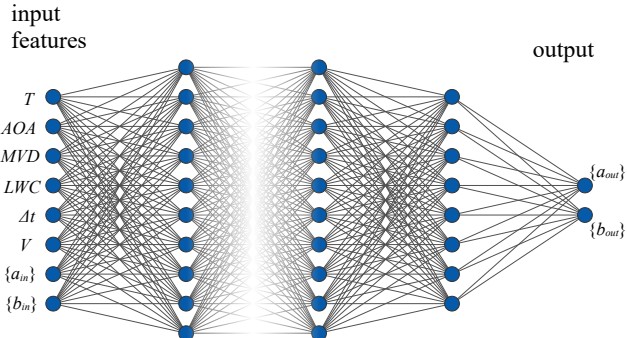

**Figure 5.** FCNN architecture.

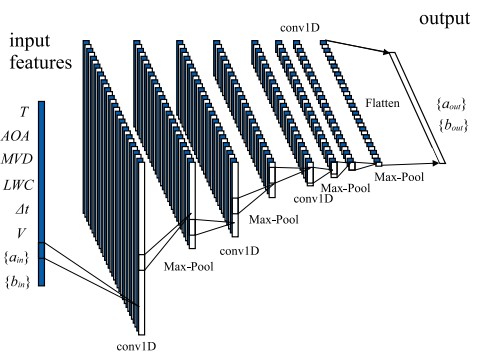

**Figure 6.** CNN architecture.

In the past few years, convolutional neural networks have been actively used to study climate. Examples of using CNN include publications on the prediction of SIC (Sea Ice Concentration) and SIT (Sea Ice Thickness) in the Arctic and the North Seas [75]. A team of authors from the UK used a CNN modification of U-Net to develop the IceNet library. IceNet was an ensemble of 25 CNNs. The CNN architecture adopted for each member of the IceNet ensemble was a U-Net neural network. U-Net is a CNN encoder–decoder in which a network feature extraction encoding path downsamples input data followed by a decoding path that upsamples the data.

The neural network was trained on initial data calculated from the dynamic climate model of the European Center for Medium-Range Weather Forecasts (ECMWF). The IceNet neural network was trained to predict monthly mean sea ice concentration for the next 6 months on 25 km maps based on climate modeling data spanning 1850–2100 and observations from 1979 to 2011.

IceNet has outperformed the leading physical model ECMWF in seasonal forecasts of sea ice conditions in the Arctic, especially for extreme summer ice events.

CNN was used in the publication of authors from the University of Utrecht, the Netherlands [76]. Using the CNN-LSTM model, predictions of the shape of ice in the Arctic, in the Barents Sea, were made using satellite data and a weather forecast model for up to several months.

CNN was used to predict SIC and SIT in the Barents and Kara Seas in [77]. The spatial scope of this study was the Arctic Ocean region (180° W–180° E/40° N–90° N), and the temporal coverage was 30 years between 1988 and 2017.

CNN has been used to solve computational mechanics problems. For example, to predict the shape of the flame from the results of the calculation data by the solver in OpenFOAM [78], to predict the dynamic modes in the model combustion chamber [79] and to predict the structure of the flame in the cavity according to the experimental data [80].

The deep-architecture-based methods (CNN and MLP) show better performance on the test set than the shallow one (SLFNN) and traditional machine learning methods (kNN

and SVM). The proposed CNN-based method achieves the highest accuracy on the test set and shows excellent capacities of feature extraction and generalization.

Recently, a convolutional neural network (CNN)-based autoencoders (AE) has also widely been used thanks to the concept of filter sharing in CNN, which enables us to handle high-dimensional fluid datasets efficiently.

The CNN-AE and the POD were applied to a wake of NACA0012 airfoil to reduce the dimension and examined the temporal behavior in the latent space [81].

A customized CNN-AE was applied to extract and visualize the nonlinear AE modes by considering a two-dimensional periodic cylinder wake and its transient [82].

They demonstrated that the CNN-AE is similar to POD but a single nonlinear AE mode contains multiple POD modes thanks to the nonlinear activation function. A similar idea has also been extended in paper [83], utilizing a hierarchical CNN-AE to present ordered AE modes following the energy contributions aiming at more efficient and interpretable compression of turbulence [84].

CNN has been utilized in image processing and classification tasks. Moreover, the use of CNN has also emerged in the fluid dynamics field because of the compatibility of the filter sharing idea to high-dimensional fluid data.

The convolutional neural network was used to predict the vortex structure of the fluid flow in the wake of a cylinder [83].

To improve the prediction performance, several neural networks have been tested along with their modifications. Besides using two types of neural networks, FCNN and CNN, and applying them with two different loss functions, see Section 4.5, two additional methodologies: batch normalization and neurons dropout, are used. Thus, there are 12 NN: $\mathrm{FCNN_{IoU}}$, $\mathrm{FCNN_{IoU}^{B}}$, $\mathrm{FCNN_{IoU}^{B,D}}$, $\mathrm{FCNN_{MSE}}$, $\mathrm{FCNN_{MSE}^{B}}$, $\mathrm{FCNN_{MSE}^{B,D}}$, see Table 1 and $\mathrm{CNN_{IoU}}$, $\mathrm{CNN_{IoU}^{B}}$, $\mathrm{CNN_{IoU}^{B,D}}$, $\mathrm{CNN_{MSE}}$, $\mathrm{CNN_{MSE}^{B}}$, $\mathrm{CNN_{MSE}^{B,D}}$, see Table 2. Here a lower index means the error type (MSE and IoU for mean squared error and intersection over union respectively), and the upper index denotes whether the model applies a batch normalization layer to every output neuron ($B$) or drops out 50% of output neurons ($D$).

**Table 1.** Architecture for different FCNN models.

| Layer Type | Output Shape | Number of Trainable Parameters | FCNN | FCNN $^{B}$ | FCNN $^{B,D}$ |
|---|---|---|---|---|---|
| Dense | 64 | 3136 | contains | contains | contains |
| Batch Normalisation | 64 | 256 | - | contains | contains |
| Activation (ReLu) | 64 | 0 | - | contains | contains |
| Dropout 50% (ReLu) | 64 | 0 | - | - | contains |
| Dense | 128 | 8320 | contains | contains | contains |
| Batch Normalisation | 128 | 512 | - | contains | contains |
| Activation (ReLu) | 128 | 0 | - | contains | contains |
| Dropout 50% (ReLu) | 128 | 0 | - | - | contains |
| Dense | 256 | 33,024 | contains | contains | contains |
| Batch Normalisation | 256 | 1024 | - | contains | contains |
| Activation (ReLu) | 256 | 0 | - | contains | contains |
| Dropout 50% (ReLu) | 256 | 0 | - | - | contains |
| Dense | 128 | 32,896 | contains | contains | contains |
| Batch Normalisation | 128 | 512 | - | contains | contains |
| Activation (ReLu) | 128 | 0 | - | contains | contains |
| Dropout 50% (ReLu) | 128 | 0 | - | - | contains |
| Dense | 64 | 8256 | contains | contains | contains |
| Batch Normalisation | 64 | 256 | - | contains | contains |
| Activation (ReLu) | 64 | 0 | - | contains | contains |
| Dropout 50% (ReLu) | 64 | 0 | - | - | contains |
| Dense | 41 | 2665 | contains | contains | contains |
| Total parameters: | | | 88,393 | 89,673 | 89,673 |

**Table 2.** Architecture for different CNN models.

| Layer Type | Output Shape | Number of Trainable Parameters | CNN | CNN [B] | CNN [B,D] |
|---|---|---|---|---|---|
| Batch Normalisation | 48 | 4 | contains | contains | contains |
| Activation (ReLu) | 48 | 0 | contains | contains | contains |
| Convolution | $45 \times 32$ | 160 | contains | contains | contains |
| Batch Normalisation | $45 \times 32$ | 128 | - | contains | contains |
| Activation (ReLu) | $45 \times 32$ | 0 | - | contains | contains |
| Dropout 50% (ReLu) | $45 \times 32$ | 0 | - | - | contains |
| Convolution | $42 \times 32$ | 4128 | contains | contains | contains |
| Batch Normalisation | $42 \times 32$ | 128 | - | contains | contains |
| Activation (ReLu) | $42 \times 32$ | 0 | - | contains | contains |
| Dropout 50% (ReLu) | $42 \times 32$ | 0 | - | - | contains |
| Convolution | $39 \times 32$ | 4128 | contains | contains | contains |
| Batch Normalisation | $39 \times 32$ | 128 | - | contains | contains |
| Activation (ReLu) | $39 \times 32$ | 0 | - | contains | contains |
| Dropout 50% (ReLu) | $39 \times 32$ | 0 | - | - | contains |
| Convolution | $36 \times 32$ | 4128 | contains | contains | contains |
| Batch Normalisation | $36 \times 32$ | 128 | - | contains | contains |
| Activation (ReLu) | $36 \times 32$ | 0 | - | contains | contains |
| Dropout 50% (ReLu) | $36 \times 32$ | 0 | - | - | contains |
| Convolution | $30 \times 32$ | 4128 | contains | contains | contains |
| Batch Normalisation | $30 \times 32$ | 128 | - | contains | contains |
| Activation (ReLu) | $30 \times 32$ | 0 | - | contains | contains |
| Dropout 50% (ReLu) | $30 \times 32$ | 0 | - | - | contains |
| Convolution | $27 \times 32$ | 4128 | contains | contains | contains |
| Batch Normalisation | $27 \times 32$ | 128 | - | contains | contains |
| Activation (ReLu) | $27 \times 32$ | 0 | - | contains | contains |
| Dropout 50% (ReLu) | $27 \times 32$ | 0 | - | - | contains |
| Convolution | $24 \times 32$ | 4128 | contains | contains | contains |
| Batch Normalisation | $24 \times 32$ | 128 | - | contains | contains |
| Activation (ReLu) | $24 \times 32$ | 0 | - | contains | contains |
| Dropout 50% (ReLu) | $24 \times 32$ | 0 | - | - | contains |
| Pooling | $12 \times 32$ | 0 | contains | contains | contains |
| Flatten | 384 | 0 | contains | contains | contains |
| Dense | 41 | 15,785 | contains | contains | contains |
| Total parameters: | | | 44,543 | 45,355 | 45,355 |

The main aim of model training is empirical risk minimization—calibrating the coefficients of the model so as to minimize the average error on the training sample [85]. The first attempts to build approximation models based only on dense layers for FCNN or using filters and pooling layers for CNN led to strong overfitting. This is clear from the high spread of errors in the training and test samples (see results for $FCNN_{IoU}$, $FCNN_{MSE}$, $CNN_{IoU}$ and $CNN_{MSE}$, Figures 7 and 8). This NN does not give the desired performance. These results are due to a high variance of training data or excess model complexity. On the other hand, simplifying the model entails the other extreme is underfitting and results between these two architectures are unsatisfied.

Figures 7 and 8 demonstrate the comparison of model performance for two metrics. To compare two sets of models the MAE error was computed, see Figure 9.

To solve the overfitting problem one can use approaches such as batch normalization or regularization with dropout. The neural network design guidelines state that the batch normalization layer must follow the fully connected layer prior to activation [86].

In Figure 9 one can see that batch normalization layers significantly improved results and decrease the test error for the IoU metric, but not for MSE. One of the possible reason for this is that the MSE is an absolute error, while IoU is relative. That means that the model trains faster and there is overfitting even for models with batch normalization and dropout.

The next modification of the NN is adding *D* layers, which drops out 50% of neurons. These models demonstrate better performance, see Figure 9.

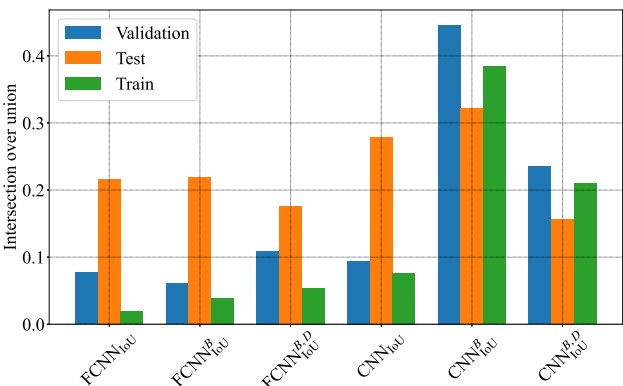

**Figure 7.** Comparison of the losses in the IoU metric for different neural network modifications.

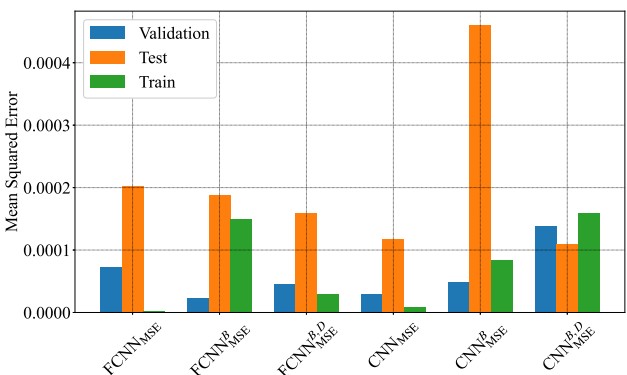

**Figure 8.** Comparison of the losses in the MSE metric for different neural network modifications.

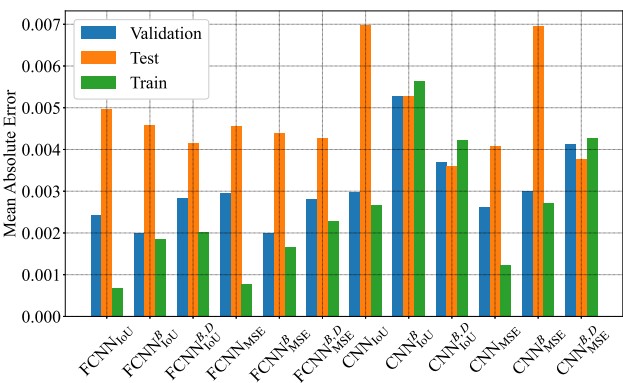

**Figure 9.** Comparison of the losses in the MAE metric for different neural network modifications.

## 5. Results

According to the MAE metric, Figure 9, one can see that the convolutional neural network with batch normalization and dropout layers demonstrates the best performance. Moreover, the results in Figure 9 demonstrate that there is almost no difference between the metrics IoU and MSE.

To train the models, the cases (see Table A1) were randomly divided into three groups:

- Train: (15 cases) 421, 73,195.02, 642, 645, 424, 128, 422, 423, 425, 214, 625, 622, 629, 632, 122;
- Validation: (3 cases): 129, 621, 124;
- Test: (4 cases): 407, 613, 222, 626.

The number of cases ratio for datasets is 68%/14%/18% for Training/Validation/ Testingrespectively. The total number of pairs in format {*input*, *target*} is 147,679.

The results for FCNN$_{IoU}^{B,D}$ and FCNN$_{IoU}^{B,D}$ are presented in Figure 10–13.

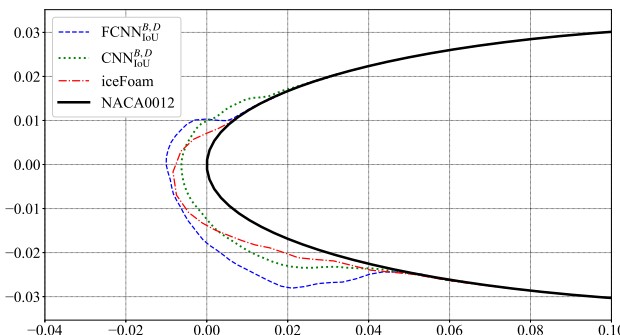

**Figure 10.** Comparison of the NACA0012 407 case ice shape for iceFoam, FCNN$_{IoU}^{B,D}$ and CNN$_{IoU}^{B,D}$.

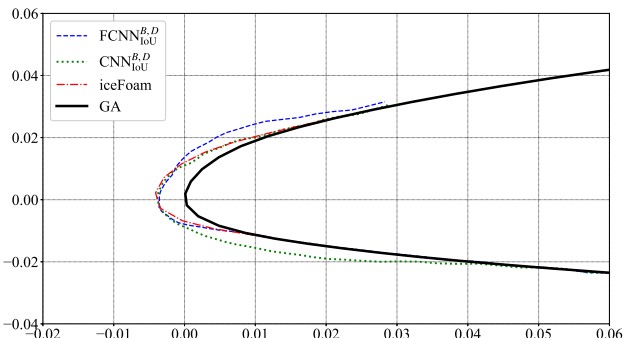

**Figure 11.** Comparison of the GA 613 case ice shape for iceFoam, FCNN$_{IoU}^{B,D}$ and CNN$_{IoU}^{B,D}$.

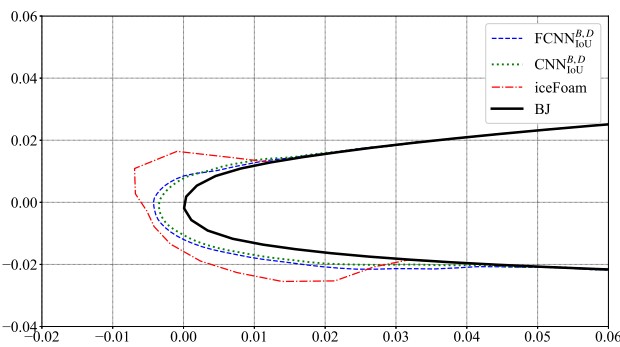

**Figure 12.** Comparison of the BJ 222 case ice shape for iceFoam, FCNN$_{IoU}^{B,D}$ and CNN$_{IoU}^{B,D}$.

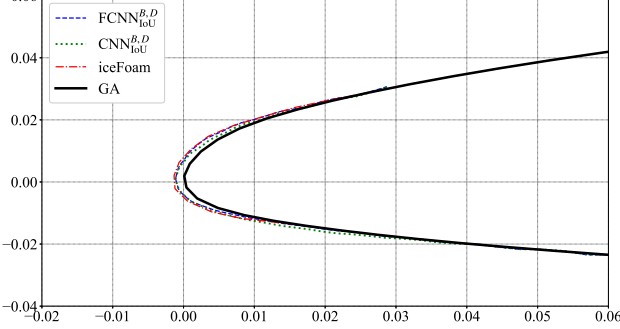

**Figure 13.** Comparison of the GA 626 case ice shape for iceFoam, FCNN$_{IoU}^{B,D}$ and CNN$_{IoU}^{B,D}$.

Figure 14 demonstrates the dynamic of losses on the training and validation datasets over training epochs.

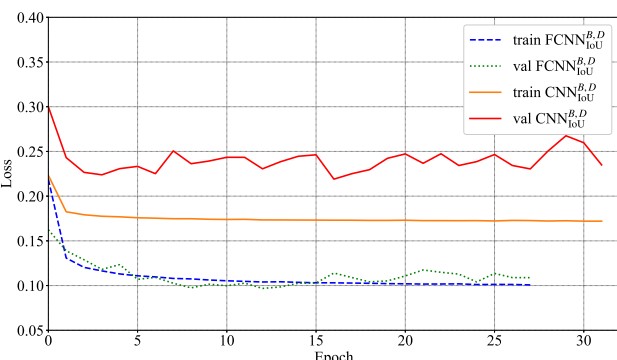

**Figure 14.** Loss per epoch for $FCNN_{IoU}^{B,D}$ and $CNN_{IoU}^{B,D}$.

Compared to the CFD packages' computational time, which is from several hours to a few days, the neural network prediction process takes about 3–5 min, including time for training.

The final result of the presented work is the specific library development for ice shape prediction using neural networks—iceMPLNet.

## 6. Discussion

It is worth paying attention to Figures 7–9, which portray error function comparison for different neural networks and metrics. The results for all FCNN variations, as well as for pure CNN, (without additions) are quite predictable. For CNN and FCNN, regardless of the chosen metric for a pure neural network, one can observe the effect of overfitting: an extremely small error on the training set, more on the validation set, and an order of magnitude more on the test set. In the case of FCNN, the overfitting effect slightly decreases with the introduction of batch normalization and is absent for models with batch normalization and dropout, which one can see in Figure 9. However, the cases of CNNs to which batch normalization and dropout are applied look somewhat anomalous (strange). In these cases, the error on the test set is less than on the validation and training sets, which are of the same order. This result can be explained by the fact that the use of normalization and dropout methods is controversial regarding CNN; in addition, there are various combinations of these methods, including those in which their joint use is excluded. On the other hand, such results in Figures 7–9 which could mean that there was an insufficient number of examples and their quality (which was described above), as a result of which the effect of network undertraining appears. It is also possible that a revision of the CNN neural network architecture is necessary. Nevertheless, the authors considered it important to present these results here. More detailed research on the application of CNNs for ice formation forecasting will be carried out in further work.

The main reasons for the difference between the results of the neural network and iceFoam are the data scarcity and the heterogeneity of ice formation regimes. The machine learning model reproduces one of the modes well, while adapting worse to the other. To solve this problem in the future it is planned to use clustering, i.e., identifying characteristic ice formation modes and using its machine learning model for each regime.

Numerical modeling of glaze ice and mixed ice regimes requires consideration of many parameters. For example, in the case of glaze ice, a water film is formed, which can accumulate, flow down the airfoil, and freeze, forming a complex ice shape, and also partially detach from the airfoil. At present, the iceFoam solver allows cosnideration of various icing parameters; however, not all features of the described physical process are included in the model. For example, droplet separation is not taken into account, the heat transfer coefficient model (htc) needs to be improved, and only one film model has been implemented. Nevertheless, the obtained numerical results for the rime ice regime

and glaze ice (for some cases) are in good agreement with the experiment and with the simulations results of other numerical packages.

The computational cases presented in Figures 10–13 are examples of the glaze ice mode, so the numerical simulation by the iceFoam solver strongly depends on the heat transfer coefficient and the film model, which also needs to be taken into account. It is assumed that neural networks overtrain a certain pattern and then poorly reproduce ice formation in another case. Insufficient numerical calculations were carried out in the glaze ice mode, as a result of which the neural network was undertrained to predict the shape of ice in this mode. It is also possible that, for some training cases, the ice shape is determined by the structure and detail of the mesh, which can also affect the results.

The presented results indicate that machine learning for ice prediction based on numerical calculations is a promising area for research. ML has a significant advantage over numerical simulation—speed, but accuracy is still a weak point. Within the framework of this work, it was possible to achieve a qualitative agreement between the results. However, a strong deviation of the parameters from the training sample can give unpredictable results and probably incorrect ones. Therefore, the applicability of the presented model is limited. To improve the accuracy of forecasts, it is necessary to significantly rework the training output, carry out additional calculations, and upgrade iceFoam.

It is important to note that changing training, validation, and test set configuration does not significantly affect the numerical results. There are a lot of possible combinations, but it was decided to focus on the case that represents the most representative cases for various airfoils. However, the authors agree that the particular configuration may influence the results obtained. In this case, it is necessary to note the great heterogeneity of the presented computational cases for training, since all these cases are taken from real experiments [71,87], which is the main limitation. As the results showed, there were not enough data for training, despite the method of increasing the training sample—data augmentation. This is also due to the limitation of the range of experimental values (specific values of temperature, velocity, angles of attack, etc.), as well as the fact that not all experiments from [71,87] were considered (due to the complexity of individual cases).

Perhaps, in this case, it would be more optimal to use a homogeneous grid of computational cases with a given step of changing parameters and the formation of a kind of decision space, based on which a neural network could be trained. However, such work incurs large computational and time costs. The authors consider this work as one of the promising directions for the future.

The iceFoam solver, developed to calculate the icing process of a 3D airfoil, was used in this work in a 2D formulation for comparison with experimental data. Important factors that determine the accuracy of ice surface calculations are the water film model and the value of the heat transfer coefficient when convective heat transfer is taken into account. We believe that the SWIM model is quite successful. In particular, it provides a natural process to calculate the flow of water over the film with its subsequent freezing in those parts of the airfoil where drops do not fall at all. A rather primitive formula was used to calculate the heat transfer coefficient. However, for the needs of the tasks being solved, it showed quite satisfactory results. The empirical spatial scale was calculated using the simple formula $L = C \times 0.75 \times 10^{-3}$, where $C$ is the airfoil chord length. In the future, it is necessary to develop a model of convective heat transfer taking into account the roughness of ice.

Future research directions are to study the effects of icing on different parts of 3D aircraft wings. The future development of iceMPLNet is aimed to ice mass and main aerodynamic coefficient prediction. Another possible direction of iceMMLNet development is "Transfer Learning", which could allow modeling of 3D wings icing using neural networks trained on 2D cases.

## 7. Conclusions

The iceFoam solver has been developed as part of the OpenFOAM package to simulate the process of airfoil icing. Its features are the use of the Euler–Lagrangian approach to describe the behavior of the gas-droplet flow, the possibility of using various models of the water film on the ice surface, a dynamic computational grid, and the possibility of using parallel calculations. The solver allows us to flexibly implement various approaches and algorithms (film models, mesh movement, turbulence models, particle behavior models) without additional programming or, in extreme cases, using classical object-oriented programming.

The ice accretion flow was simulated using the iceFoam solver, the unsteady RANS mathematical model with the $k$-$\omega$ SST turbulence model and two different neural networks. The error in the computation of the ice thickness on the cylinder and airfoils with rime ice was less than 5%.

The initial datasets were generated based on data for four different airfoils and initial physical values of flow. The airfoil shape of ice was represented with a function using parabolic coordinate system's transformation. For the function approximation the corresponding Fourier series is used. As a result, Fourier coefficients were received which were selected as features. The approach with augmentation of data was used to enhance the volume of data using data from intermediate time moments.

In this study, a framework of a deep-learning-based solver for ice accretion simulation was established, which was realized by integrating an optimized deep convolutional neural network, iceMPLNet. A case with a 2D airfoil in gas-droplet flow was used as an example to demonstrate the performance of the proposed method in terms of both accuracy and efficiency.

By training FCNN and CNN with different metrics and data from the flow around an airfoil at several different parameters (inlet velocities, LWC, diameter of droplets, $t_{ice}$, angle of attack), CNN demonstrated the feasibility of simulating results both within and even beyond the range of the inlet velocity of training data.

Our approach included some new features, such as training a neural network on various airfoils, using all time slices as input for learning. Several neural networks with batch normalization and dropout layers are also used. The intersection over union (IoU) error $err_{\text{IoU}}$ is used, which is the ratio of the ice shape difference area and the total area of the predicted and target ice shapes.

In terms of efficiency, CNN is characterized by bringing the huge amount of computation forward to the training phase, while reducing the cost of simulations hundred-fold.

To the best of the author's knowledge, this is the first application of deep learning to reconstruct ice accretion using a combined approach with CNN and iceFoam. Generally speaking, our study can be seen as a proof of concept, demonstrating that the prediction of turbulent flow and species distributions can be reformulated as a machine learning problem. The application of this framework will make real-time numerical simulation possible, thus accelerating the development of technologies such as digital twins, rapid prototyping with applications to complex systems such as aircraft and wind turbines. The robustness and versatility of the proposed CNN framework will be further tested for a wide range of ice accretion problems in follow-on research with "Transfer Learning".

The neural network was defined as FCNN with five hidden layers and CNN with pooling layers. The datasets were split into train, validation and test parts. The two error metrics were defined for both neural networks.

The most accurate result of ice shape prediction is received with FCNN and CNN that applied batch normalization and dropped out 50% of layers to output neurons of each layer.

**Author Contributions:** Conceptualization, K.K. and D.R.; methodology, D.R. and K.K.; validation, D.R. and S.S.; formal analysis, S.S.; investigation, S.S.; resources, A.I.; data curation, D.R. and A.I.; writing—original draft preparation, K.K., S.S. and D.R.; writing—review and editing, D.R.; visualization, A.I.; supervision, S.S.; project administration, K.K.; funding acquisition, S.S. All authors have read and agreed to the published version of the manuscript.

**Funding:** This research was supported by the Ministry of Science and Higher Education of the Russian Federation, agreement No. 075-15-2020-808.

**Institutional Review Board Statement:** Not applicable.

**Informed Consent Statement:** Not applicable.

**Data Availability Statement:** The data presented in this study are available on request from the corresponding author.

**Acknowledgments:** We acknowledge support of our colleagues Valeria Melnikova, Andrey Osipov. This work was carried out using high-performance computing resources of the federal center for collective usage at the NRC Kurchatov Institute, http://ckp.nrcki.ru/ (accessed on 1 February 2022). The iceMPLNet library is named in honor of our colleague, a remarkable scientist and a wonderful person—Mikhail Petrovich Levin (30.06.1955–12.01.2021). He took part in projects related to icing and in many other works of our laboratory.

**Conflicts of Interest:** The authors declare no conflict of interest.

## Abbreviations

The following abbreviations are used in this manuscript:

| | |
|---|---|
| NN | Neural network |
| ANN | Artificial neural network |
| FCNN | Fully connected neural network |
| CNN | Convolutional neural network |
| LWC | Liquid water content |
| MVD | Median volume diameter |
| AOA | Angle of attack |
| CFD | Computational Fluid Dynamics |
| ROM | Reduced Order Modeling |
| POD | Proper Orthogonal Decomposition |
| RANS | Reynolds-averaged Navier–Stokes |
| GAMG | geometric-algebraic multi-grid |
| SWIM | Shallow Water Icing Model |
| MSE | Mean squared error |
| MAE | Mean absolute error |
| IoU | Intersection over Union |
| ISP RAS | Institute for System Programming of the Russian Academy of Sciences |
| ONERA | Office National d'Etudes et de Recherches Aérospatiales |
| CIRA | Centro Italiano Ricerche Aerospaziali |
| TUDA | Technical University of Darmstadt |
| TUBS | Technical University of Braunschweig |
| PoliMI | Politecnico di Milano |

## Appendix A

Appendix A contains details and data of supplemental flow physical parameters for selected airfoils. All cases from Table A1 are taken from NASA experiments, described in works [71,87].

**Table A1.** This is a table caption of physical parameters for airfoils.

| Number | Airfoil | T (K) | AOA. | LWC (g/m$^3$) | MVD | t (min) | V, (m/s) |
|---|---|---|---|---|---|---|---|
| 122 | CT | 263.65 | 1.6 | 0.563 | 21 | 4.9 | 130.15 |
| 124 | CT | 263.65 | 0.7 | 0.563 | 21 | 4.9 | 130.15 |
| 128 | CT | 258.55 | 0.6 | 0.341 | 21 | 2 | 128.6 |
| 129 | CT | 263.65 | 0.7 | 0.563 | 21 | 2 | 130.15 |
| 214 | BJ | 262.61 | 6 | 0.6 | 15 | 6 | 90 |
| 222 | BJ | 262.5 | 1.5 | 0.43 | 20 | 6 | 128.6 |
| 407 | NACA0012 | 256.32 | 4 | 0.4 | 20 | 9.8 | 102.8 |
| 421 | NACA0012 | 268.4 | 4 | 1 | 20 | 6 | 67.1 |
| 422 | NACA0012 | 266.74 | 4 | 1 | 20 | 6 | 67.1 |
| 423 | NACA0012 | 265.07 | 4 | 1 | 20 | 6 | 67.1 |
| 424 | NACA0012 | 259.51 | 4 | 1 | 20 | 6 | 67.1 |
| 425 | NACA0012 | 244.51 | 4 | 0.9 | 20 | 6 | 67.1 |
| 613 | GA | 263.15 | −0.2 | 0.56 | 15 | 6 | 92.6 |
| 621 | GA | 268.15 | 1.9 | 0.54 | 20 | 2 | 66.87 |
| 622 | GA | 268.15 | 1.8 | 0.54 | 20 | 6 | 66.87 |
| 625 | GA | 263.15 | 1.8 | 0.66 | 40 | 0.9 | 66.87 |
| 626 | GA | 263.15 | 0.3 | 0.44 | 20 | 2 | 66.87 |
| 629 | GA | 258.15 | 0.3 | 0.44 | 20 | 1.4 | 66.87 |
| 632 | GA | 263.15 | 0.3 | 0.6 | 15 | 2 | 66.87 |
| 642 | GA | 263.15 | −1.7 | 0.44 | 20 | 5.9 | 66.87 |
| 645 | GA | 263.15 | 2.4 | 0.44 | 20 | 5.9 | 66.87 |
| 73,195.02 | BJ | 258.15 | 1.5 | 0.31 | 20 | 5.8 | 129 |

CT—Commercial Transport airfoil, BJ—Business Jet airfoil, NACA—National Advisory Committee for Aeronautics, GA—General Aviation airfoil. T—Temperature, AOA—Angle of Attack, MVD—Median Volume Diameter, LWC—Liquid Water Content, t—Time of Ice Accretion, V—Velocity.

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
