# Peer review of "Neural Network Prediction for Ice Shapes on Airfoils Using iceFoam Simulations"

_aerospace, doi:10.3390/aerospace9020096_

Round 1
Reviewer 1 Report
This paper entitled "Neural Network Prediction for Ice Shapes on Airfoils Using iceFoam Simulations" deals with methods for the prediction of ice shapes based on artificial neural networks (ANNs). The first introductory part proposes an exhaustive state of the art of the codes dedicated to icing simulation. The authors also propose a review of the methods for data and input parameters management. The second part presents the different models used to calculate the ice shapes. These models are implemented in the OpenFOAM package. Section 3 proposes a comparison and an assessment among the ice shapes obtained with different solvers, including the one proposed by the authors. Two reference cases are studied (RUN421 and RUN425) based on a NACA0012 profile. Section 4 is the one that describes the core and the novelties of the method presented in this paper. By changing to a parabolic coordinate system, the airfoil profile is represented by a set of Fourier coefficients. The weights of the Fourier series are parameters for the neural network. Then, the architectures of the neural networks are described. Note that a novelty lies in the fact that all time slices are used as inputs for the learning step. Two loss functions are proposed: Mean Squared Error (MSE) and Intersection over Union (IoU). Convolutional Neural Networks (CNN) and Fully Connected Neural Networks (FCNN) are compared. Two additional methodologies, batch normalization (B) and neurons dropout (D) are proposed. Part 5 is about results where different combinations among FCNN vs. CNN, MSE vs. IoU for the loss functions and (B) vs. (D) methods are proposed. The three data sets for training, validation and tests are presented. A brief discussion is provided in Section 6. Finally conclusions are drawn.
The objectives of this paper are clearly identified, namely the use of artificial intelligence (AI) tools for ice shape prediction. The proposed approach attempts to address this challenge. Before it can be published in Aerospace, the article must be improved in a few ways:
1) Discussion part (Sec. 6) is a bit brief. In particular, if we focus on figures 10, 11 and 12, the results between the calculations obtained with an AI tool and those from the classical iceFoam approach are quite different. On the other hand, the comparison is very good for figure 13. How do you explain this ? Why are neural networks good for some configurations and not so good for others ? Are there any trends in favor of using AI ? What happens if, for the validation test cases, you deviate a lot from the configurations used for training ? All these questions should be discussed.
2) The English is to be revised for the whole article. This is a major point and a main weakness of the article.
3) Typos: Eq. (23), shouldn't xi and yi be rather xi' and yi' ?
For all these reasons, I recommend a minor revision for this article.
Reviewer 2 Report
In this manuscript, icing shape prediction methods were examined by FCNN and CNN based on numerical simulations of airfoil icing by iceFoam. The primary guidance of the neural network and numerical simulation method of iceFoam were denoted. Then, it is explained that the ice formation is predicted by using CNN machine learning based on the results obtained from iceFoam. However, machine learning such as CNN is an already existing method, and I considered that if the possibility of predicting ice formation is to be clarified, it would be better to discuss the prediction of ice formation and its accuracy for objects that deviate from the training conditions. In addition, the argument that CNNs can and cannot be used for prediction is an ad-hoc argument, and its universality and robustness should be carefully examined.
Therefore, I do not recommend the present paper for publication in Aerospace.
Reviewer 3 Report
Review comments
Title: Neural Network Prediction for Ice Shapes on Airfoils Using iceFoam Simulations
Manuscript Number: aerospace-1480930
General comments:
This manuscript presents a study of the development of artificial neural networks (ANNs) for the ice accretion prediction for different airfoils. Many important parameters and equations in aircraft icing simulations are also introduced in great detail. The paper is interesting and provides some values in terms of aircraft icing predictions. However, there are several points of criticism that will require a major revision by the authors before the publication can be considered.
Detailed comments:
- The discussion of cable/wire icing problems in the introduction part is less related to do with the research topic in this paper. The authors should consider deleting this section.
- In the section of Mathematical model for ice accretion simulation, the authors should make sure to include all references for the equations/theories used.
- The authors mentioned that “The main assumption of the SWIM model is that the film temperature is equal to the temperature of the triple point of water”. Please justify.
- In the calculation of the heat transfer coefficient, traditional empirical equations are used. The author should provide a detailed discussion of how they consider the effects of ice roughness, and the coupled effects of heat convection and flow interactions.
- When discussing the numerical simulations, the authors should be very careful about presenting the different parameters. For example, what is the reason for selecting the specific domain sizes? What is case 421? How is the cell number determined?
- Please give the definition of Courant number.
- The reviewer is confused about the illustration of Equation (27). Please give more details on how the equation is derived.
- Overall, the English writing should be significantly improved.
Round 2
Reviewer 2 Report
The paper has been greatly improved. However, there are too many line breaks, so please reconsider the paragraph structure.
Reviewer 3 Report
Thank you for considering my comments and revising the paper. It is now good to publish.